# Preparation of Duplex Sequencing Libraries for Archival Paraffin-Embedded Tissue Samples Using Single-Strand-Specific Nuclease P1

**DOI:** 10.3390/ijms23094586

**Published:** 2022-04-21

**Authors:** Natalia V. Mitiushkina, Grigory A. Yanus, Ekatherina Sh. Kuligina, Tatiana A. Laidus, Alexandr A. Romanko, Maksim M. Kholmatov, Alexandr O. Ivantsov, Svetlana N. Aleksakhina, Evgeny N. Imyanitov

**Affiliations:** 1Department of Tumor Growth Biology, N.N. Petrov Institute of Oncology, 197758 St.-Petersburg, Russia; nmmail@inbox.ru (N.V.M.); octavedoctor@yandex.ru (G.A.Y.); kate.kuligina@gmail.com (E.S.K.); tanyxax@yandex.ru (T.A.L.); romanko.aleksandr.a@gmail.com (A.A.R.); maksim.holmatov@gmail.com (M.M.K.); ivantsovalexandr81@gmail.com (A.O.I.); pamparam24@gmail.com (S.N.A.); 2Department of Medical Genetics, St.-Petersburg Pediatric Medical University, 194100 St.-Petersburg, Russia; 3Department of Oncology, I.I. Mechnikov North-Western Medical University, 191015 St.-Petersburg, Russia

**Keywords:** duplex sequencing, FFPE, BotSeqS, nuclease P1, colorectal carcinoma, tumor mutation load

## Abstract

DNA from formalin-fixed paraffin-embedded (FFPE) tissues, which are frequently utilized in cancer research, is significantly affected by chemical degradation. It was suggested that approaches that are based on duplex sequencing can significantly improve the accuracy of mutation detection in FFPE-derived DNA. However, the original duplex sequencing method cannot be utilized for the analysis of formalin-fixed paraffin-embedded (FFPE) tissues, as FFPE DNA contains an excessive number of damaged bases, and these lesions are converted to false double-strand nucleotide substitutions during polymerase-driven DNA end repair process. To resolve this drawback, we replaced DNA polymerase by a single strand-specific nuclease P1. Nuclease P1 was shown to efficiently remove RNA from DNA preparations, to fragment the FFPE-derived DNA and to remove 5′/3′-overhangs. To assess the performance of duplex sequencing-based methods in FFPE-derived DNA, we constructed the Bottleneck Sequencing System (BotSeqS) libraries from five colorectal carcinomas (CRCs) using either DNA polymerase or nuclease P1. As expected, the number of identified mutations was approximately an order of magnitude higher in libraries prepared with DNA polymerase vs. nuclease P1 (626 ± 167/Mb vs. 75 ± 37/Mb, paired *t*-test *p*-value 0.003). Furthermore, the use of nuclease P1 but not polymerase-driven DNA end repair allowed a reliable discrimination between CRC tumors with and without hypermutator phenotypes. The utility of newly developed modification was validated in the collection of 17 CRCs and 5 adjacent normal tissues. Nuclease P1 can be recommended for the use in duplex sequencing library preparation from FFPE-derived DNA.

## 1. Introduction

Methods that are based on duplex sequencing [1] are known to be extremely sensitive and have a number of different applications. In particular, they can be used to identify underrepresented genetic changes and mutations, which constitute only a minor percentage of a studied sample [1,2,3]. Moreover, duplex sequencing can facilitate studies of mutagenic effects of chemical compounds by providing the possibility of detecting somatic mosaic mutations arising in laboratory animals or cell cultures [4]. A special method, the Bottleneck Sequencing System (BotSeqS), was developed to analyze rare somatic mutations in normal human tissues [5].

The idea of duplex sequencing-based approaches relies on the double-stranded nature of DNA molecules. Mutations that are present within living cells usually affect both DNA strands. On the contrary, artificial mutations that arise in vitro due to chemical modification of DNA bases, mistakes made by DNA polymerase during PCR amplification or wrong sequencer readings can only be found in one DNA strand and its copies, because they are not accompanied by the change of a complementary nucleotide in the opposite DNA sequence. Thus, it is possible to discriminate between true and artificial DNA alterations by comparing the sequences of two DNA strands of a particular DNA fragment.

The error frequency of duplex sequencing is estimated to be lower than 1 × 10^−9^ [1]. However, it was recognized recently that the originally suggested approach was not as accurate as it could be due to the use of DNA polymerase in the end repair process during the preparation of the DNA library [2,6]. While filling in 5′-overhangs, DNA polymerase can ‘copy’ errors, which initially were present in a single strand only, to the second DNA strand, and such errors then cannot be corrected by duplex sequencing. New approaches were suggested to alleviate this problem, which include the use of a single-strand-specific nuclease to blunt DNA ends or fragmenting DNA with an enzyme, which does not leave sticky ends [7]. The indicated trouble is especially pronounced when using nucleic acids isolated from formalin-fixed paraffin-embedded (FFPE) tissues. FFPE samples are commonly utilized in cancer research and diagnostic routine as the most available source of tumor DNA, but unfortunately, DNA extracted from FFPE tissues is known to be severely degraded, and NGS sequencing of such DNA may result in a high rate of false mutations due to multiple artifacts [8]. It was suggested that approaches that are based on duplex sequencing can significantly improve the accuracy of mutation detection in FFPE-derived DNA [8,9]. Here, we applied one of these approaches, the BotSeqS method [5], for the study of somatic mutations frequency in colorectal tumor samples stored as archival FFPE tissue blocks.

To address the problem associated with the incorporation of false mutations during the DNA end repair process, we decided to substitute DNA polymerase with a single-strand-specific nuclease. Single strand-specific nucleases are multifunctional enzymes capable of hydrolyzing phosphodiester bonds in RNA, single-stranded DNA (ssDNA) and in single-stranded regions of double-stranded nucleic acids [10]. Interestingly, one of these enzymes, S1 nuclease, has already been used in DNA library preparations from FFPE tissues in two studies. In one of them, S1 nuclease was shown to effectively shear FFPE-derived DNA, producing DNA fragments in an optimal size range for subsequent NGS analyses [11]. Indeed, DNA from FFPE tissue contains enough nicks and gaps, which can be used as cutting sites by a single-strand-specific nuclease. Cutting DNA molecules within already damaged sites may be advantageous, because otherwise, these sites may appear inside library fragments, preventing them from being amplified and sequenced. Another study utilized the same nuclease S1 to remove ssDNA before library preparation, which resulted in the improved quality of NGS results [12]. In both of these studies, the treatment with nuclease S1 was followed by the standard DNA end repair process. It still remains to be determined which nuclease would be the most suitable for the preparation of duplex sequencing libraries from FFPE-derived DNA and how well treatment with a single-strand-specific nuclease can replace processes such as end repair or DNA fragmentation. We attempted to clarify these questions in the current study.

## 2. Results

### 2.1. Choosing Enzyme to Replace DNA Polymerase in the End Repair Process

The summary of the experimental design implemented in this study is shown in Figure 1. We started from choosing the enzyme capable of replacing DNA polymerase in the end repair step during DNA library preparation. First, we compared three commercially available single-strand-specific nucleases, mung bean nuclease (MBN), nuclease S1 and nuclease P1, in their ability to blunt end DNA for subsequent ligation. For this purpose, DNA was extracted from blood lymphocytes of two healthy individuals. After enzymatic shearing, 1000 ng aliquots of these DNA samples were treated, each by one of the single-strand-specific nucleases, followed by T4 polynucleotide kinase. One aliquot of each sample was also subjected to a standard end repair procedure, while another one was only phosphorylated with T4 polynucleotide kinase to serve as a control. The same Y-adapters were ligated to all samples and concentrations of the obtained libraries were measured by real-time qPCR. Among the studied enzymes, the best results were obtained with the use of nuclease P1, independently of its concentration in reaction (10 U or 100 U) (Figure 2a). We found that treatment with MBN under conditions recommended by the manufacturer did not result in a significant increase in the amount of fragments with ligated adapters relative to control samples, while nuclease S1 demonstrated moderate ability to blunt end DNA. Although, among studied enzymes, nuclease P1 showed the highest efficiency in our experiments, the standard end-repair procedure outperformed it, resulting in almost twice higher library concentrations (Figure 2a). Further analysis of libraries by capillary electrophoresis did not reveal significant differences in sizes between differently treated samples (Appendix A).

Next, we repeated the above experiments with two samples of tumor DNA extracted from FFPE material. Single strand-specific nucleases are known to act both on RNA and single-stranded DNA. As both samples contained a significant amount of RNA, for each experiment we took an aliquot that contained 1000 ng of DNA plus RNA. These samples purposefully were not subjected to DNA fragmentation, because we expected FFPE-derived DNA to be fragmented by single-strand-specific nucleases, as it was shown by Chun et al. [11]. First, we measured the amount of RNA in samples before and after treatment with nucleases. In accordance with previous results, treatment with MBN did not result in a reduction of RNA amount with respect to controls, nuclease S1 showed moderate activity toward RNA, while no detectable RNA remained in samples treated with nuclease P1 (Appendix A). Then, after the ligation of adapters, libraries were measured by real-time PCR. The highest concentrations of libraries, again, were found in samples treated with nuclease P1 (Figure 2b). Somewhat lower concentrations were obtained with conventional end repair in these experiments, as was expected, because of a lack of prior fragmentation. The high amount of adapter dimers was found by capillary electrophoresis in samples treated by MBN or nuclease S1, as well as in the control samples and one of the samples subjected to standard end repair procedures (Figure 2b, Appendix A); adapter dimers were formed most likely because the amount of DNA fragments suitable for adapter ligation was limited in these samples.

Finally, we sought to explore if the additional fragmentation of FFPE-derived DNA, conducted prior to nuclease P1 treatment, could help to increase library concentrations. To address this question, we subjected two other aliquots of FFPE samples to short enzymatic fragmentation, followed by either treatment with nuclease P1 and T4 polynucleotide kinase or the conventional end repair process. The results are shown in Figure 2b and Appendix A. While fragmentation clearly improved the results of the library preparation from FFPE-derived DNA when using conventional end repair, it did not result in further advantages for samples treated with nuclease P1. Thus, we concluded that nuclease P1 could be used to both share and blunt end FFPE-derived DNA, and no prior fragmentation is necessary for this type of material.

### 2.2. Comparing the DNA Libraries Prepared from FFPE Samples Using Standard and Nuclease P1-Based Method

To assess the suitability of FFPE-derived DNA for the analysis by duplex-sequencing based methods, we have chosen the Bottleneck Sequencing System (BotSeqS) method developed by Hoang et al. [5]. This approach was developed to quantify rare somatic mutations in normal tissues; it involves the analysis of a rather limited number of randomly selected DNA fragments for each sample, which makes it economically attractive. The drawback of the method is the requirement of whole-genome sequencing to obtain information on germ-line variants in sequenced samples. However, as the number of somatic mutations in colorectal carcinomas, especially in those with DNA repair defects, is significantly greater than in normal tissues, we decided that it could be sufficient to use population genomics database data and only filter out variants with higher than 1% population frequency. We suggested that this would still allow us to distinguish tumors with strong hypermutator phenotype and to illustrate the benefits of using nuclease P1 in duplex sequencing library preparation from FFPE tissues. BotSeqS protocol requires the ligation of Y-adaptors to DNA fragments, which is followed by PCR amplification of only a limited amount of initial DNA fragments (‘bottlenecking’ step), usually covering significantly less than one full genome sequence. As a consequence, genomic coordinates of the mapped sequences can serve as unique identifiers of each sequenced DNA molecule.

We prepared BotSeqS libraries from five colorectal carcinoma FFPE samples, either using nuclease P1-based fragmentation and end repair or using conventional DNA fragmentation and end repair methods. Two of the samples were known to have microsatellite instability (MSI), and another one was from a patient with hereditary mutation in POLD1 gene. After the ligation of adaptors, libraries were measured using digital droplet PCR, and approximately 20,000 random DNA fragments from each library were taken in PCR amplification and NGS sequencing. After filtering ambiguously mapped reads, reads with soft-clipped bases and improper read pairs, two consensus sequences for each strand of the DNA molecules were constructed by combining data from at least five PCR copies using custom python scripts (see Materials and Methods for details). Finally, these sequences were merged so that the resulting duplex sequence contained only those nucleotides, which were supported by both consensus sequences.

While the overall number of reads per sample was not significantly different between nuclease P1 and standard DNA libraries, the percentage of reads that passed filters was higher in libraries prepared with nuclease P1, which was probably attributed to the higher median template length in these libraries (154 ± 19 bp vs. 121 ± 11 bp, paired t-test *p*-value 0.004). As expected, the number of duplex sequencing-supported mutations that remained after filtering out the common SNPs was significantly higher in libraries prepared with the conventional end-repair method (626 ± 167/Mb vs. 75 ± 37/Mb, paired t-test *p*-value 0.003). Moreover, in these libraries, mutations occurred more frequently toward the ends of the template sequence, while no such tendency was observed in libraries prepared using nuclease P1 (Figure 3). Notably, the mutational frequency was two times greater in MSI-positive tumors than in wild-type or POLD1-positive tumors when counted in libraries prepared with nuclease P1 (115 ± 15/Mb vs. 53 ± 6/Mb and 39/Mb, respectively), while no such difference was observed in libraries prepared using the standard approach (579 ± 181/Mb vs. 720 ± 220/Mb and 533, respectively).

### 2.3. Quantifying the Number of Mutations in NGS Libraries Prepared with Nuclease P1 from Colorectal Carcinoma Samples with or without Hypermutator Phenotype

To explore further the possibility of using nuclease P1 for duplex sequencing library preparation from FFPE-derived DNA, we extended our collection of colorectal cancer cases by adding two more MSI-positive cases, three cases with germline biallelic mutations in the *MUTYH* gene, one case with hereditary *POLD1* variant and another one with somatic mutation in *POLE* gene, as well as four cases with none of these alterations (Table 1). In five cases, pathologically normal adjacent tissue was also available for analysis. It should be commented that the germ-line variants detected in *POLD1* gene were of unknown or uncertain clinical significance (Table 1). Additionally, a sample of DNA extracted from white blood cells of a healthy volunteer was subjected to the same library preparation pipeline, after initial enzymatic shearing, to serve as a control.

Main bioinformatics analysis results for the 23 BotSeqS libraries are provided in Appendix A. Bioinformatic pipeline was further optimized at this stage: reads with an excessive number of mutations were excluded, and the reads ends were trimmed, because they are frequently misaligned in regions containing insertions or deletions (see Materials and Methods for details). The overall number of read pairs obtained for each FFPE-derived DNA library varied from 1.1 to 3.3 million (mean 1.9 ± 0.6)and 64 ± 14% of these reads passed filters. However, in one tumor-normal pair, only 25 and 35% of reads passed filters, most likely due to the very short median template length (82 and 86 bp, respectively). The median template length was 146 ± 27 bp in all FFPE-derived DNA libraries, while it approached 213 bp in the library, which was prepared from blood-derived DNA. When restricting analysis to reads that passed filters, the number of unique templates in individual samples was 15156 ± 4924, which is generally consistent with approximately 20,000 library fragments selected for PCR amplification and sequencing. However, among these templates, only 18 ± 5% formed duplexes in the FFPE-derived DNA libraries, despitethe excessive number of duplicates analyzed, while in a library prepared from healthy donor’s blood DNA, 5 copies of each strand of 47% templates were successfully read. The total length of ‘duplex sequences’ per sample in the FFPE cohort ranged from 0.15 to 0.51 Mb, with 1.0 Mb of duplex sequences obtained with blood DNA sample. It should be noted that the initial number of read pairs in the blood-derived DNA library (1.4 millions) did not differ from that in the FFPE libraries in this study. The number of mutations, which affected only one of the DNA strands and were corrected by duplex sequencing, varied greatly in FFPE-derived DNA samples (range 181–2419 per Mb of duplex sequences, mean 1054 ± 723). This value was also high in the blood-derived DNA library (229 mutations/Mb). Known polymorphisms with overall population frequencies higher than 1% (as provided by gnomAD v.3 database) were found at frequencies of 809–1097 per Mb of duplex sequences in all samples (mean 967 ± 71/Mb).

The number of duplex sequencing-supported mutations in all pathologically normal adjacent tissue samples was 21 ± 8/Mb (range 6–29); the high number of such mutations (19/Mb) was found also in DNA extracted from the blood cells. Tumors from patients with hereditary mutations in MUTYH or POLD1 genes were present with a very similar number of mutations (25 ± 7 and 13 ± 9 per Mb, respectively), which was also true for the tumors with no identified germline or somatic defects in genes responsible for DNA maintenance and repair (26 ± 6 mutations per Mb) (Figure 4). The four MSI-positive and one POLE-positive tumors, however, showed clear hypermutator phenotype, with 116 ± 50 and 115 mutations per Mb, respectively. Notably, 50–82% of the mutations identified in MSI-positive samples were short insertions and deletions, while 92% of the mutations identified in the POLE-positive sample were single-nucleotide variants.

## 3. Discussion

Formalin-fixed paraffin-embedded (FFPE) tissues are routinely used in cancer genetic studies and diagnostics and are unlikely to be substituted by other type of material in the near future [8,9]. The quality of FFPE-derived DNA varies greatly depending on conditions of a sample storage and fixation parameters. Duplex sequencing-based approaches can improve the results of NGS analysis of DNA extracted from FFPE tissue. In this work, we applied one of the duplex sequencing-based methods, BotSeqS, for the study of somatic mutational loads in a collection of FFPE-derived colorectal carcinoma DNA samples, which included several cases with a known hypermutator phenotype.

DNA lesions arising during formalin fixation can lead to the misincorporation of certain nucleotides in new DNA chains synthesized by DNA polymerases during NGS library preparation. One of the most frequent sources of such errors in FFPE-derived DNA is the deamination of cytosine bases to uracil [8]. Treatment with uracil DNA glycosylase (UDG) is often recommended for improving the quality of such DNA [13]. UDG converts uracils within DNA molecules to abasic sites, significantly diminishing the DNA polymerase extension rate of the respective templates. However, in the current study, we refrained from using UDG; we aimed to have more DNA chains amplified and sequenced given that errors affecting only a single chain of the DNA duplex are normally corrected by the duplex sequencing technique.

Unfortunately, there are other types of lesions in DNA extracted from paraffin blocks, which are more difficult to account for. FFPE-derived DNA can be heavily fragmented; moreover, it contains nicks and gaps, which often prevent one of the two DNA strands from being PCR-amplified and sequenced. As duplex sequencing requires readings of both DNA strands, this could lead to a significant decrease in the total number of obtained duplex sequences and, ultimately, to the analysis failure. In view of this, the use of single-strand-specific nucleases to fragment FFPE-derived DNA can provide an advantage over standard methods of DNA shearing, as these enzymes specifically cut DNA at nicks and gaps, preventing these lesions from occurring within DNA library fragments.

The weakness of the original duplex sequenced methodology stems from the use of DNA polymerase in the end repair process: By filling in 5′-overhangs and single stranded gaps in the DNA, this enzyme ‘copies’ mistakes that were initially present in one DNA strand to the other strand of the same molecule (Figure 5); such mistakes then cannot be corrected. As it was demonstrated in our study, when FFPE-derived DNA is subjected to duplex sequencing with standard polymerase-based DNA end repair, a large number of mutations can be found (Figure 3). In recent studies, attempts were made to improve duplex sequencing results by replacing DNA polymerases with single-strand-specific nucleases in the end repair process. In particular, pretreatment with S1 nuclease in the study of Otsubo et al. [14] was shown to increase the accuracy of duplex sequencing, while less satisfactory results were obtained with mung bean nuclease and RecJf exonuclease. It should be noted that following sonication and nuclease pretreatment, DNA fragments were also subjected to conventional DNA repair in the mentioned study; thus, it was not possible to assess directly the ability of nuclease S1 to blunt-end DNA. On the contrary, Abascal et al. [7] completely substituted DNA polymerase with mung bean nuclease in the end repair process; this was undertaken upon the development of a modification of the BotSeqS method, named NanoSeq. Unfortunately, the publication of Abascal et al. [7] does not provide information regarding actual mung bean nuclease performance. It should be noted that the BotSeqS method does not require all DNA sample to be successfully converted to adapter-ligated library fragments, because, normally, less than one full genomic sequence needs to be covered [5]. With sufficient amounts of high-quality DNA, it is not unlikely that BotSeqS/NanoSeq libraries can be produced even without the DNA end repair step.

In the current study, we compared the ability of three commercially available enzymes (mung bean nuclease, S1 nuclease and nuclease P1) to (1) remove RNA from DNA preparations, (2) to remove ‘blunt end’ DNA and (3) to fragment FFPE-derived DNA. Under the conditions recommended by the manufacturers, nuclease P1 performed best in all conducted experiments, while significantly poorer results were obtained with S1 nuclease. Surprisingly, the mung bean nuclease demonstrated an apparent lack of activity in all analyses. The results did not change after the enzyme was reordered from the same manufacturer (data not shown). It is possible, however, that the performance of the mung bean nuclease and S1 nuclease could be improved by optimizing the parameters of the reactions. Notably, although Abascal et al. [7] also used mung bean nuclease manufactured by New England BioLabs (Ipswich, MA, USA), as in the current study, the reaction was set up differently: only 50 ng of sheared DNA was taken into reaction with 1 U or less of the enzyme in a total volume of 30 mkl [7]. Chun et al. [11] and Otsubo et al. [14] used S1 nuclease provided by the other manufacturer (Promega Corporation, Madison, WI, USA) [11,14]. The lack of data on the performance of the same enzymes provided by different manufacturers and under the conditions that differ from the ‘default’ protocols may be recognized as a limitation of the current investigation. Another limitation is the use of only a single tumor type, colorectal carcinoma, in our study, as it was reported that DNA extracted from different types of archival tumor and normal tissues can vary significantly in quantity and quality [15].

Using the selected enzyme, nuclease P1, to fragment and blunt end DNA, we successfully prepared and sequenced 22 BotSeqS libraries from FFPE-derived tumor and normal DNA samples. Although we did not have whole-genome sequencing data and, thus, could not discriminate between somatic and rare germline variants found in our samples, we were able to correctly identify colorectal carcinoma samples with the most pronounced hypermutator phenotypes, specifically, MSI-positive and *POLE* mutation-positive cases. However, no difference was found when *MUTYH*- and *POLD1*-positive tumor and normal DNA samples were compared to tumor and normal DNA samples from patients with no hereditary cancer-predisposing mutations or MSI. This is generally consistent with the results of Viel et al. [16], who determined mean somatic mutation frequencies in colorectal carcinomas from patients with *MUTYH*-associated polyposis to be as low as 5.3/Mb [16]. On the contrary, the literature data suggest that high tumor mutational burden is characteristic for tumors with defects in *POLE*/*POLD1* genes [17,18,19]. However, it should be noted that germline alterations in *POLD1* gene in patients included in the current study were of unknown or uncertain functional significance. This can explain the lack of increase in mutation frequency in *POLD1*-positive cases. The BotSeqS method in the form presented in this article could not be recommended for the general assessment of mutational burden in tumors due to the high background error rate, which is likely related to both non-accounted rare germline variants and the errors that appear due to the lack of reads realignment in insertion/deletion regions. However, more accurate approaches could be developed on the basis of duplex sequencing methodologies that have the potential to improve genotyping and tumor mutation burden evaluation in FFPE tumor samples.

Although the current study demonstrated the general applicability of the duplex sequencing-based methods for the study of FFPE-derived DNA samples, it should be noted, that the percentage of duplexes (DNA fragments with both forward and reverse strands successfully analyzed) was lower in these samples in comparison to what was reported previously for high quality DNA samples [5] or relative to the control blood-derived DNA sample used in this study. Consequently, significantly more DNA fragments are needed to be subjected to NGS analysis in order to obtain the necessary coverage with duplex sequences when high-quality blood or fresh-frozen tissue DNA is substituted with FFPE-derived DNA.

In conclusion, this study confirms that FFPE-derived DNA can by studied by duplex sequencing-based methods, although the effectiveness of analysis may be low in some samples due to the presence of multiple lesions that prevents the successful amplification of both DNA strands. Nuclease P1 can be used to both fragment and blunt end DNA extracted from FFPE tissues.

## 4. Materials and Methods

### 4.1. Samples and DNA Extraction

Colorectal carcinoma FFPE samples from 17 patients treated in the years 2013–2019 were obtained from the archive of N.N. Petrov Institute of Oncology (St. Petersburg, Russia). The investigation was conducted in accordance with the Helsinki Declaration and followed the rules of the local Ethics Committee. Nucleic acids from FFPE tissues were extracted with Quick-DNA/RNA MagBead kit (Zymo Research, Irvine, CA, USA) according to the manufacturer’s instruction, and the amount of DNA and RNA was measured with Qubit 4.0 fluorometer (Thermo Fisher Scientific, Waltham, MA, USA). DNA from the blood of two healthy volunteers was extracted using the salt–chloroform method, as described elsewhere [20].

### 4.2. Preparations of NGS Libraries from FFPE- and Blood-Derived DNA Using Standard and Single-Strand-Specific Nuclease-Based Methods

NEBNext dsDNA Fragmentase (New England BioLabs, Ipswich, MA, USA) was used to shear DNA according to the manufacturer’s instruction. The time of shearing for the DNA extracted from blood was set to 20 min. Wherever indicated, FFPE-derived DNA was sheared for 5 min. After enzymatic fragmentation and following all other enzymatic reactions that preceded the ligation of adaptors, DNA fragments were purified with MinElute PCR Purification Kit (Qiagen, Hilden, Germany); this helped us minimize DNA loss and reduce the volumes of the subsequent reactions, as compared to the purification on magnetic beads.

For the treatment with nucleases, 1000 ng of sheared blood DNA or 1000 ng of DNA plus RNA, extracted from FFPE tissues, was used per reaction. Mung bean nuclease (MBN; New England BioLabs, Ipswich, MA, USA) treatment was performed as recommended by the manufacturer. Briefly, nucleic acids were dissolved in 10 μL of 1X mung bean Nuclease Reaction Buffer and 1 unit of MBN was added per reaction. Incubation at 30℃ lasted for 30 min, and after that, SDS was added to a 0.01% concentration. Reaction with S1 Nuclease (Thermo Fisher Scientific, Waltham, MA, USA) contained 6 μL of 5X Reaction Buffer for S1 Nuclease, 0.1 μL (10 U) of the enzyme and nuclease-free water relative to the total volume of 30 μL. Incubation was carried out for 30 min at room temperature and stopped by the addition of 2 μL of 0.5 M EDTA and heating at 70℃ for 10 min. For treatment with Nuclease P1 (New England BioLabs, Ipswich, MA, USA), 1X NEBuffer 1.1 was used instead of Nuclease P1 Reaction buffer to limit nuclease’s activity toward dsDNA, as recommended by the manufacturer. Two different concentrations of the enzyme were tried, as there was no single recommended concentration for this enzyme. The reaction volume was 20 μL, and incubation was carried out at 37℃ for 30 min before the addition of 2 μL of 0.5 M EDTA to stop the reaction. Following treatment with nucleases, T4 Polynucleotide Kinase (New England BioLabs, Ipswich, MA, USA) was used to phosphorylate 5′ DNA ends. The standard end repair process was conducted with NEBNext End Repair module (New England BioLabs, Ipswich, MA, USA) [21].

Klenow Fragment (3′- > 5′ exo-) (New England BioLabs, Ipswich, MA, USA) was used for the A-tailing, following the manufacturer’s protocol. Y-adapters for Illumina sequencing were ligated to all samples as follows: 1.2 μL of adapters was mixed with 3.8 μL of column-purified A-tailed DNA fragments; subsequently, 5 μL of Blunt/TA Ligase Master Mix (New England BioLabs, Ipswich, MA, USA) was added to each sample. Reaction was incubated at 25℃ for 30 min. For the initial experiments, a universal SynoDuplex adapter (25 μM), prepared as described by Ren et al. [3], was ligated to all samples. However, it was substituted with Illumina adapters (Truseq DNA Single Indexes Sets A/B, Illumina, San Diego, CA, USA) in the construction of libraries for actual NGS analysis. Full-length adapters allowed us to quantify NGS libraries more precisely with a digital droplet PCR (ddPCR). Following adapter ligation, samples were cleaned with 0.9X AmPure magnetic beads (Beckman Coulter, Brea, CA, USA), and all libraries were eluted in 25 mkl volume of 10mM Tris-HCl (pH 8.0).

### 4.3. Quantification of DNA Libraries by Real-Time PCR

Real-time qPCR reactions for the quantification of libraries were conducted using the CFX96 Real-Time PCR Detection System (Bio-Rad, Hercules, CA, USA). Each reaction contained 1X GeneAmp PCR buffer I (Applied Biosystems, Waltham, MA, USA), 0.05 U/μL Taq M hot-start DNA polymerase (AlkorBio, St.-Petersburg, Russia), 2.5 mM MgCl2, 250 μM of each dNTP, 1-x solution of EvaGreen dye (Biotium, Fremont, CA, USA) and 200 nM of each primer (5′-CAAGCAGAAGACGGCATACGAGATATTGGCGTGACTGGAGTTCAGACGTGT-3′ and 5′-AATGATACGGCGACCACCGAGATCTACACTCTTTCCCTACACGAC-3′). The volume of PCR reaction was 20 μL, and 2 μL of 1:10000 diluted DNA library was added per reaction. Libraries were quantified against the sequential dilutions of two 2 nM libraries, which were prepared earlier from FFPE materials. The dilution process and qPCR measurements were repeated twice for each library, and the mean value of two measurements was taken as the final result. Fragment sizes of amplified libraries were measured with 5200 Fragment Analyzer System (Agilent Technologies, Santa Clara, CA, USA).

### 4.4. Preparation of BotSeqS Libraries

The following primers and TaqMan probes were used for the quantification of libraries using QX100 Droplet Digital PCR System (Bio-Rad, Hercules, CA, USA): AATGATACGGCGACCACCGA, CAAGCAGAAGACGGCATACG, R6G-ACACGTCTGAACTCCAGTCAC-BHQ1, FAM-TCTACACTCTTTCCCTACACGA-BHQ1. The PCR reactions were set up according to the protocol, supplied by the manufacturer. All droplets positive for at least one dye were considered in the calculation of the libraries’ concentration.

Libraries that were considered for NGS sequencing were PCR-amplified using Q5 High-Fidelity DNA Polymerase (New England BioLabs, Ipswich, MA, USA) with the following primers: 5′-AATGATACGGCGACCACCGAGATCTACACTCTTTCCCTACACG*A-3′ and 5′-CAAGCAGAAGACGGCATACGAGA*T-3′, where * indicates the phosphorothioate bond. The composition of each PCR reaction included 1X Q5 Reaction buffer, 200 μM of each dNTP, 0.5 μM of each primer, 0.02 U/μL DNA polymerase, the appropriate amount of DNA library (20,000 fragments with ligated adaptors, as quantified by ddPCR) and nuclease-free water. The reaction was carried out in 50 μL volume with the indicated cycling conditions: 98 ℃ 30 s, then 30 cycles of 98 ℃ 15 s, 60 ℃ 30 s and 65 ℃ 45 s, followed by 65 ℃ for 5 min. PCR-amplified libraries were cleaned with 0.9X AmPure magnetic beads, measured by Qubit fluorometer and Agilent 5200 Fragment Analyzer System, pooled and sequenced on the Illumina MiSeq instrument (Illumina, San Diego, CA, USA) using MiSeq Reagent Kit v2 (300-cycles) in the paired reads mode.

### 4.5. Data Analysis

The full bioinformatic analysis pipeline is provided in Appendix A. The following programs and resources were utilized: Burrows-Wheeler Aligner (BWA v.0.7.17) [22], SAMtools v.1.9 [23], Picard Toolkit v.2.18.14 [24], Genome Reference Consortium Human Build 38 (GRCh38) [25] and The Genome Aggregation Database (gnomAD) v.3 [26]. The custom python scripts, which we used in bioinformatic analysis, were based primarily on the *pysam* package [27]. Statistical data analysis was performed with the R software [28], *stats* package; the functions of the *graphics* and *ggplot2* [29] R packages were used for plotting.

## Figures and Tables

**Figure 1 ijms-23-04586-f001:**
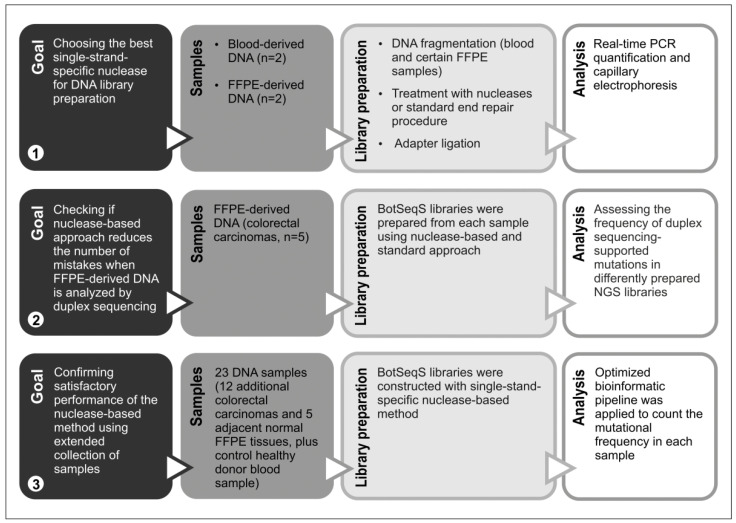
General overview of the experiments described in the article.

**Figure 2 ijms-23-04586-f002:**
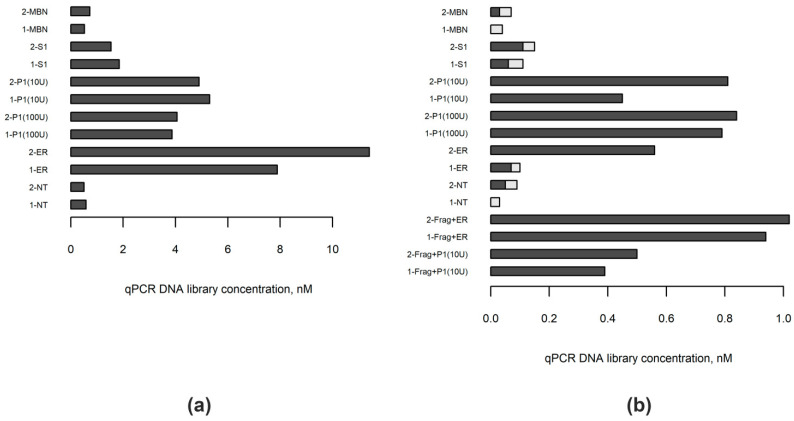
Concentrations of libraries prepared from blood (*n* = 2) and FFPE-derived (*n* = 2) DNA samples with different enzymes utilized to repair DNA ends. (**a**) Separate aliquots of blood DNA samples 1 and 2 were enzymatically fragmented, then the ends of the resulting DNA fragments were blunted either with one of the single-strand-specific nucleases (mung bean nuclease, MBN; S1 nuclease, S1; nuclease P1, P1) or by using the standard polymerase-based approach (ER). One aliquot of each DNA sample was left untreated (NT). (**b**) FFPE-derived DNA samples 1 and 2 were treated similarly; however, only two aliquots of each sample were subjected to enzymatic fragmentation (Frag) before proceeding to blunt DNA ends step. The concentration of fragments with successfully ligated adaptors in each library was measured by quantitative real-time PCR (qPCR). The proportions of adapter dimers were calculated on the basis of library fragment size distribution analysis and are indicated in the picture as light grey-colored parts of the bars.

**Figure 3 ijms-23-04586-f003:**
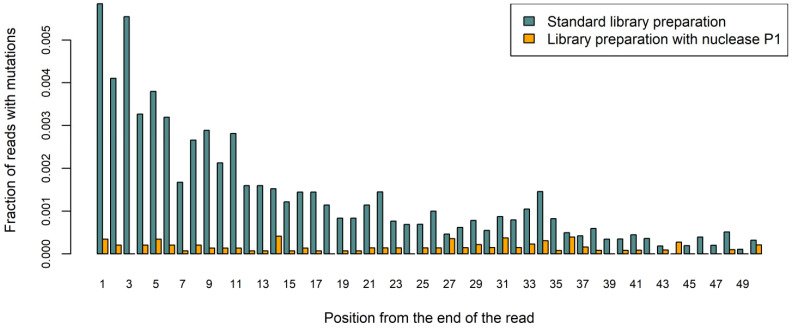
A higher number of duplex sequencing-supported mutations occurs toward the ends of DNA fragments in libraries prepared with a common approach, which utilizes DNA polymerase to blunt-end DNA, but not in libraries prepared with the nuclease P1-based method. NGS libraries were prepared from five colorectal carcinoma FFPE-derived DNA samples, and aggregated data are shown in the figure. Positions from 1 to 50 from either end of each DNA fragment were analyzed.

**Figure 4 ijms-23-04586-f004:**
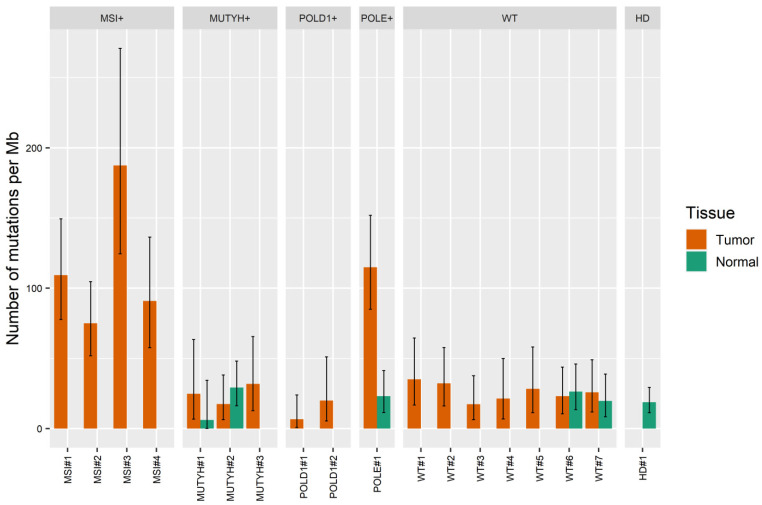
Number of mutations per Mb of duplex sequences found in colorectal carcinomas samples and healthy donor DNA with modified Bottleneck Sequencing System (BotSeqS) method. The 95% confidence intervals were calculated for each sample using the binomial distribution.

**Figure 5 ijms-23-04586-f005:**
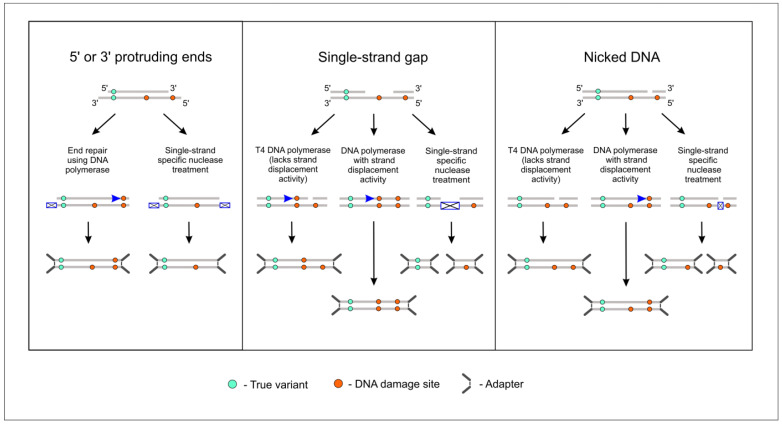
Use of DNA polymerases to repair DNA ends before NGS adapter ligation leads to the incorporation of errors in the synthesized complementary DNA strand, which is opposite to the damaged sites, when filling in 5′-protruding ends or single-strand gaps, or when strand resynthesis is initiated from nicks in DNA. Substitution of DNA polymerases by single-strand-specific nucleases helps avoid this problem.

**Table 1 ijms-23-04586-t001:** Characteristics of colorectal carcinoma cases studied with the BotSeqS method.

Label	Sex	Age	Main Genotyping Result	Mutations in *KRAS*, *NRAS* or *BRAF* Genes
MSI#1	female	35	MSI	No
MSI#2	female	71	MSI	*BRAF* V600E
MSI#3	female	45	MSI	No
MSI#4	female	78	MSI	*KRAS* Q61K
MUTYH#1	female	48	*MUTYH* Y179C (ClinVar VCV000005293.37, pathogenic), G396D (ClinVar VCV000005294, pathogenic)	*KRAS* G12C
MUTYH#2	female	59	*MUTYH* Y179C (ClinVar VCV000005293.37, pathogenic), G396D (ClinVar VCV000005294, pathogenic)	*KRAS* G12C
MUTYH#3	male	39	*MUTYH* R245H (ClinVar VCV000140877.19, pathogenic), G396D (ClinVar VCV000005294, pathogenic)	*KRAS* G12C
POLD1#1	male	42	*POLD1* A516V (dbSNP rs752755096, rare SNP, significance unknown)	*BRAF* V600E
POLD1#2	female	51	*POLD1* V538I (ClinVar VCV001025450.1, uncertain significance)	*BRAF* V600E
POLE#1	male	33	*POLE* S297F (somatic mutation, COSMIC COSM937330, FATHMM prediction: pathogenic)	*KRAS* A146T
WT#1	male	68		*KRAS* G13D
WT#2	male	67		No
WT#3	male	68		No
WT#4	female	66		*KRAS* G12V
WT#5	female	72		*KRAS* G12V
WT#6	female	40		No
WT#7	male	69		No

## Data Availability

The data that support the findings of this study are available from the corresponding author upon reasonable request.

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
