# Peer review of "Preparation of Duplex Sequencing Libraries for Archival Paraffin-Embedded Tissue Samples Using Single-Strand-Specific Nuclease P1"

_ijms, 2022, doi:10.3390/ijms23094586_

Round 1

Reviewer 1 Report

This is an interesting manuscript describing the use of nuclease P1 to prepare DNA from parafin-embedded tissues for next generation sequencing.

I found this manuscript to be well-written and to give credit to previous work done by other authors. One could argue that the contribution to our knowledge is incremental, given that others have proposed using nuclease S1, which is also a single-stranded nuclease like nuclease P1 used by the authors in this manuscript. However, even though the contribution might be incremental, it can nevertheless be important. Thus, I recommend publication of this manuscript. I don't have specific points that the authors need to address. The manuscript can be published essentially as is.

Author Response

Reviewer has not raised critical comments.

Reviewer 2 Report

This manuscript describes the results of duplex DNA sequencing using different methods and sample types. Library preparation using mung bean nuclease, S1 nuclease, and single-strand specific P1 nuclease were compared. P1 nuclease-based procedures showed efficient RNA removal, optimal DNA fragmentation, and removal of overhangs. The number of reads which passed filters doubled. The P1 nuclease treatment of DNA removed the need for the final DNA repair process in library preparation.

The method was applied to duplex DNA sequencing from blood cells, colorectal cancer samples, and normal adjacent tissue. Libraries prepared with the P1 nuclease showed a significantly lower mutation rate than standard duplex sequencing procedures. The P1 nuclease-based method correctly identified colorectal cancer with a hypermutating phenotype in some colorectal cancer cases.

Comments

The authors provides compelling reasons for applying the P1 nuclease in NGS studies using DNA from FFPE tissues.

To validate the method described in the manuscript and establish that this method represents the best strategy for resolving artefacts in NGS sequencing of DNA from FFPE tumor tissues, the authors should extend the study to a larger number of colorectal cancer cases with or without hypermutator phenotype.

In line 51, 1 superscript has no apparent reason.

Author Response

Comment: To validate the method described in the manuscript and establish that this method represents the best strategy for resolving artifacts in NGS sequencing of DNA from FFPE tumor tissues, the authors should extend the study to a larger number of colorectal cancer cases with or without hypermutator phenotype.

Response: We agree that the number of cases analyzed in this report is small, although sufficient to show the promise of the suggested approach. This manuscript describes just a proof-of-concept study. We are planning to utilize several tumor types with varying degrees of hypermutator phenotype in the next investigation, which will be specifically designed to address the issue raised by the Reviewer.  

Comment: In line 51, 1 superscript has no apparent reason.

Response: We apologize for this error, we have corrected it.